# Pregnancy loss and its predictors among ever-pregnant women in Sub-Saharan Africa: Multilevel mixed effect negative binomial regression

**Abel Endawkie**[1]*, **Yawkal Tsega**[2]

**1** Department of Epidemiology and Biostatistics, School of Public Health, College of Medicine and Health Sciences, Wollo University, Dessie, Ethiopia, **2** Department of Health System and Management School of Public Health, College of Medicine and Health Sciences, Wollo University, Dessie, Ethiopia

* abelendawkie@gmail.com

## Abstract

### Background

Pregnancy loss is a significant maternal health issue in Sub-Saharan Africa. Africa has the highest rates of stillbirths globally, with an estimated 2.7 million stillbirths occurring each year on the continent. The pregnancy loss data are underreported and inconsistently recorded in Sub-Saharan Africa. Therefore, this study aimed to determine the number of pregnancy loss and its predictors among ever-pregnant women in Sub-Saharan Africa using a recent round of demographic and health survey (DHS) data.

### Method

A secondary data analysis was conducted among 235,086 weighted ever-pregnant women in Sub-Saharan Africa using a recent round of DHS data from 2015-2023. Multilevel mixed effect negative binomial regression was conducted. An adjusted incidence rate ratio (AIRR) with a 95% confidence interval (CI) was reported.

### Result

The median number of pregnancy loss in Sub-Saharan Africa is 2.67, 95%CI (2.64, 2.69). A one-year increase in maternal age [AIRR= 1.05, 95%CI (1.06, 1.07)], primary educational status of the mother [AIRR = 1.10, 95% CI (1.01, 1.22)], women with a partner who has higher education [AIRR= 1.18, 95% CI (1.04, 1.39)], a higher number of under-five children [AIRR =0.95, 95% CI (0.91,0.99)], women have ever pregnant in Cote'divore [AIRR 1.76, 95% CI (1.6, 2)] are associated with the number of pregnancy loss.

### Conclusion

The findings indicate that there are three pregnancy losses among ever-pregnant women in Sub-Saharan Africa. Notably, a one-year increase in maternal age and higher education levels for both mothers and their partners are linked to an increased risk of pregnancy

**Data availability statement:** Third party data was obtained for this study from The DHS Program. Data may be requested from The DHS Program after creating an account and submitting a concept note. More access information can be found on The DHS Program website (https://dhsprogram.com/data/Access-Instructions.cfm). The authors confirm that interested researchers would be able to access these data in the same manner as the authors. The authors also confirm that they had no special access privileges that others would not have.

**Funding:** The authors received no specific funding for this work.

**Competing interests:** The authors have declared that no competing interests exist.

loss. In contrast, mothers with multiple children generally experience lower rates of loss. Therefore, policy interventions should address the heightened risk of pregnancy loss linked to advancing maternal age and higher education levels for both mothers and their partners. This can be achieved by supporting programs that educate prospective parents about the effects of maternal age on pregnancy outcomes. Furthermore, promoting flexible educational pathways and providing career support can encourage healthier timing for pregnancies. Additionally, initiatives that support families and promote larger family sizes may help reduce pregnancy loss rates in Sub-Saharan Africa.

## Introduction

Pregnancy loss refers to the loss of a pregnancy before the fetus can survive outside the womb. It can also be referred to as early pregnancy loss, or mid-trimester pregnancy loss which includes abortion (spontaneous abortion/ miscarriage and induced abortion) for a pregnancy loss before 28 and stillbirth after 28 weeks of pregnancy [1–5].

Every year, approximately 2.6 million babies are stillbirths out of 136 million births in the world [6]. According to a recent report by the United Nations, one stillbirth occurs every 16 seconds, and 84% of these stillbirths occur in low and middle-income countries [7].

Between 2015 and 2020, Africa had an average of 4.44 children which was the only continent registering a fertility rate higher than the global average, which was 2.47 children per woman. In contrast to this high fertility, pregnancy loss is a significant maternal health issue in Sub-Saharan Africa. According to data from the World Health Organization (WHO), Africa has one of the highest rates of stillbirths globally, with an estimated 2.7 million stillbirths occurring each year on the continent. This means that a large number of families in Africa are affected by the loss of a baby before birth [8].

Different studies showed that pregnancy loss was affected by maternal age [9–12], educational statuses of women [9,11,13], marital status of the woman [13], husband's occupation [14], and level of education [14], media exposure [14], household wealth index [11,14], and antenatal care (ANC) visit [14] from non-clinical factors and genetic abnormalities, infections, immunological and implantation disorders, uterine and endocrine abnormalities, and lifestyle factors from clinical factors [15–17].

There are inconsistent findings in the associations of predictors with pregnancy loss. The studies conducted in Ethiopia [17,18] showed that women who attended primary and above education were less likely to have had stillbirth whereas a study conducted in Ghana [19] showed women who attended primary and above education were more likely to have had stillbirth. Similarly, there is a discrepancy in the findings regarding women who have fewer children and longer birth intervals, which may help prevent pregnancy loss [18]. However, other studies showed that fewer child numbers and long birth intervals increase pregnancy loss [20,21].

Moreover, these studies were conducted on the prevalence of pregnancy loss and recurrent pregnancy loss regardless of the number of pregnancy losses which focused on clinical factors among reproductive-age women in Sub-Saharan Africa using logistic regression analysis [10,11,13,19,22].

Grouping women who have experienced multiple pregnancy losses together with those who have had a single loss can result in a loss of important information. Families, particularly mothers, often feel profound grief from repeated losses, which can be more difficult to bear than a single loss. Therefore, conducting a count data analysis is essential to understand their experiences better. The count data analysis helps to improve information loss like

underestimation and will provide the average number of pregnancy losses a mother experienced before the survey. Therefore, this study aimed to determine the number of pregnancy losses and its predictors among ever-pregnant women in Sub-Saharan Africa. This study provides insights into both individual and contextual determinants of the average pregnancy loss a woman experiences.

## Method

### Study design and setting

Data from demographic and health surveys conducted between 2015 and 2023 among women of reproductive age who have ever been pregnant in Sub-Saharan African (SSA) countries was utilized. This study included Burkina Faso, Côte d'Ivoire, Kenya, and Tanzania, because their latest demographic and health survey (DHS) data had information on pregnancy loss (including spontaneous abortion/miscarriage, induced abortion, and stillbirth) available during this period. Countries that have no information on pregnancy loss (including spontaneous abortion/miscarriage, induced abortion, and stillbirth) on the latest DHS data were excluded from the analysis.

### Source and study population

The source population was all reproductive-age women who had a history of pregnancy before the survey in Sub-Saharan Africa, whereas those in the selected Enumeration Areas (EAs) were the study population from 2015-2023 DHS data.

### Inclusion and exclusion criteria

Those countries with women who have a history of ever pregnant before the survey were all included. Moreover, women who were pregnant and had pregnancy outcomes documented as incomplete were excluded. Women of reproductive age and infecund women were excluded. **Data source:** We extracted the dependent and independent variables from the recent DHS data of the birth record (BR) data set. The data set includes information on pregnancy, postnatal care, and children born in the last 5 years [23,24]. The DHS is a household survey that is conducted every five years in low and middle-income countries. It is a crucial data source for maternal healthcare utilization issues in these countries, as it collects data on various reproductive health issues such as marriage, pregnancy, antenatal care visits, abortion, and miscarriage [23]. The data collected from a DHS survey was organized in a hierarchical structure, with households within a cluster forming the top level. The next level consists of household members, followed by interviewed women and men as a subset of household members. The bottom levels of the hierarchy include pregnancies and children of each interviewed woman [23,24].
**Sampling method:** DHS uses a two-stage stratified cluster sampling technique. In the first stage, a sample of enumeration areas (EAs) was selected independently from each stratum with proportional allocation stratified by residence (urban and rural). In the second stage, households were taken from the selected EAs using a systematic sampling technique [24].
**Sample size**: The sample size was determined from the birth to recode file 'BR file' from the DHS dataset of SSA countries that had at least one survey conducted between 2015 and 2023 and the final sample size was 235,086 weighted reproductive age women (Fig 1) [24].
**Study variables:** The dependent variable in this study was the number of pregnancy losses that a woman had experienced before the survey, which was recorded as (0, 1, 2, 3, 4, 5, 6… 9).

The independent variables included socio-demographic and economic-related factors like (individual maternal age, educational status, marital status, religion, partner educational status

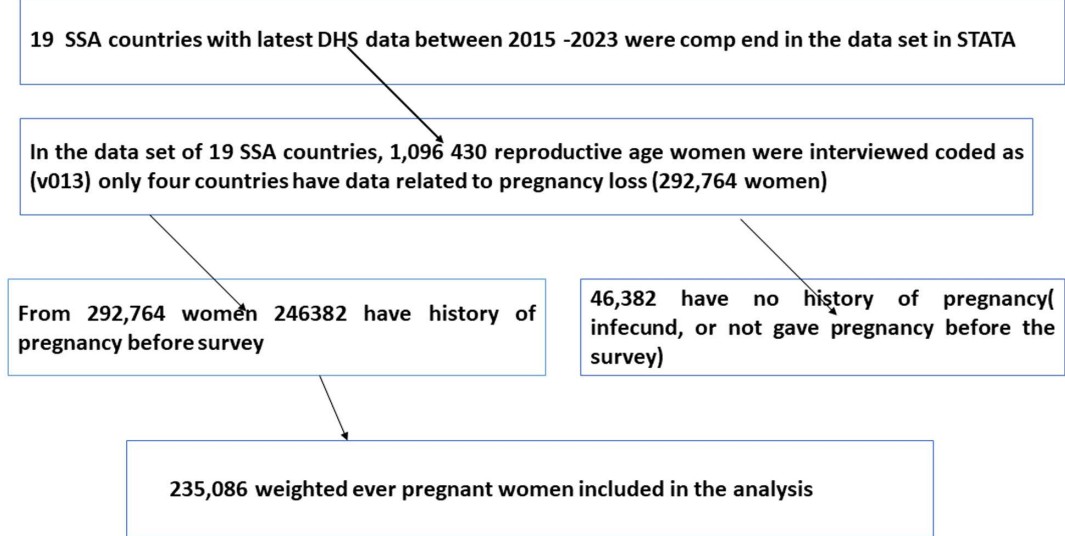

**Fig 1. The sample size determination procedure using demographic health survey data from 2015 to 2023 in sub-Saharan Africa.**

household level factors like; sex of the household head, age of the household head, household wealth index, community level factors like; residence, community level wealth index, community level educational status) and country were independent variable. It is important to note that including the country as a factor in the analysis does not imply the country itself is a causative factor for pregnancy loss. Instead, it serves as a proxy for capturing the contextual differences between countries, such as variations in healthcare infrastructure, cultural practices, socioeconomic conditions, and healthcare policies.

## Variable measurement and operational definition

**Pregnancy loss:** It includes abortion (spontaneous abortion/ miscarriage and induced abortion) for a pregnancy loss before 28 and stillbirth after 28 weeks of pregnancy [1].

The number of pregnancy losses in this study was counted as 0, 1, 2, 3….9 among reproductive-age women who ever pregnant before the survey.

**Count regression**: The count regression model was used for the non-negative integer which counted data as 0, 1, 2, 3… [25].

**Community-level variable measurement. Community-level factors:** The physical and social environments surrounding individuals, households, or families influence the probability of individuals engaging in specific behaviors? In this research, community-level factors such as community-level wealth index, community-level maternal literacy, place of residence, and country were examined.

**Community-level wealth index:** The proportion of women originating from households classified within the richest and richer wealth index categories is summed and divided by the total household wealth index value of each cluster. Those households with a wealth index value at or above the mean are classified as experiencing a high-income level, while those below the mean are categorized as having a low-income level. The mean is chosen as the cutoff point in this context due to the normal distribution observed at the economic level at the community level, as indicated by the coefficient of skewness falling between -1 and 1, signifying a normal distribution.

**Community-level maternal literacy**: The proportion of mothers who have primary and above educational status categories is summed and divided by the total maternal educational status value of each cluster. Those mothers with a maternal educational status value at or above the mean are classified as experiencing a high level of maternal literacy, while those below the mean are categorized as having a low level of maternal literacy. The mean is chosen as the cutoff point in this context due to the normal distribution observed in the level of maternal literacy at the community level, as indicated by the coefficient of skewness falling between -1 and 1, signifying a normal distribution.

**Community-level maternal ANC utilization**: The proportion of mothers who have ANC categories is divided by the total ANC utilization status value of each cluster. Those mothers with ANC utilization status values at or above the mean are classified as experiencing a high level of ANC utilization, while those below the mean are categorized as having a low level of ANC utilization. The mean is chosen as the cutoff point in this context due to the normal distribution observed in the level of media exposure at the community level, as indicated by the coefficient of skewness falling between -1 and 1, signifying a normal distribution.

**Community-level maternal media exposure**: The proportion of mothers who have media exposure categories is divided by the total media exposure status value of each cluster. Those mothers with media exposure status values at or above the mean are classified as experiencing a high level of media exposure, while those below the mean are categorized as having a low level of media exposure. The mean is chosen as the cutoff point in this context due to the normal distribution observed in the level of media exposure at the community level, as indicated by the coefficient of skewness falling between -1 and 1, signifying a normal distribution.

## Data processing and analysis

Data used were extracted, cleaned, coded, and analyzed using STATA version 17 Statistical software. Sample weights were done before further analysis, and descriptive statistics were described using frequencies, percentages, mean, and standard deviation, and presented using tables, and narratives.

**Poisson regression model.** The Poisson regression model is a statistical technique used to count data as a function of a set of independent variables. It is often used to model the relationship between a count response variable and one or more predictor variables. The model assumes that the response variable follows a Poisson distribution and that the logarithm of its expected value can be modeled as a linear combination of the predictor variables. The standard Poisson regression model assumes that the observations are independent over time and that the mean and variance of the dependent variable are equal [26]. Since the variance exceeds the mean (variance = 0.4 and mean =0.267) in this study there is over-dispersion and the premise of the Poisson regression has failed. To address the over-dispersion, the negative binomial regression model with an unobserved particular effect (random term or error term) for the parameter was chosen. The over-dispersion parameter in the negative binomial (NB) specification was tested using a Likelihood Ratio (LR) test for the parameter α (p-value < 0.001) in contrast to the Poisson model specification [25].

**Multilevel analysis:** Due to the hierarchical nature of DHS data, multilevel mixed effect negative binomial regression analysis was conducted. The intra-class correlation coefficient (ICC) was calculated using the formula[27] $ICC = \frac{\delta 2}{\delta 2 + \pi 2 / 3}$ where δ2 indicates the estimated variance of clusters and the ICC was 11.3% which was greater than 10% which favors the multilevel model analysis of negative binomial regression. Four multilevel negative binomial regression models were fitted. First, we estimated a "null" model (model 1), which merely

has a random intercept and enables us to determine whether a contextual dimension may be present for a phenomenon. Then the individual characteristics were included in the model (model 2) to investigate the extent to which the overall difference in the number of pregnancy loss was explained by the individual variation. Next, community-level variables were added to the model (model 3) to investigate whether this contextual phenomenon was conditioned by community-level characteristics. Finally, both individual and community-level characteristics were added to the model (model 4) to determine the predictors of number of pregnancy loss.

**Method of parameter estimation.** The cluster variation was evaluated using ICC. Furthermore, the overall variation was measured by the percent change of variance (PCV). PCV was calculated by following the formula.

$$PCV = \frac{\delta 2 \text{null model} - \delta 2 \text{of each model}}{\cdot 2 \text{null model}}$$

At the individual and community levels, the relationship between the explanatory variables and the number of pregnancy losses was estimated using the fixed effects (a measure of association). A p-value of less than 0.25 was used to identify potential candidates for the final model. After evaluating the adjusted incident rate ratio (AIRR) and the crude incident rate ratio (CIRR), the adjusted incident rate ratio (AIRR) was reported. A p-value of less than 0.05 was used to detect the strength of the associations between the dependent and independent variables using adjusted incident rate ratios and 95% confidence intervals. The variance inflation factor (VIF) was evaluated; the fitted model's mean VIF was 2. The best-fit model was ultimately determined by comparing the models using the information criteria.

**Ethical approval:** No ethical approval was needed because we had used the demographic and health survey which de-identifies all data before making it public, and the used DHS data sets are openly accessible. An authorization letter was requested to the Central Statistical Agency (CSA) which encompasses SSA countries, to download the DHS data set and this was obtained at https://dhsprogram.com/. The dataset and all methods of this study were conducted according to the guidelines laid down in the Declaration of Helsinki principles and based on DHS research guidelines.

## Result

### Socio-demographic characteristics study population

The study included a weighted sample of 235,086 reproductive-age women who ever pregnant before the survey and the detailed socio-demographic characteristics of respondents in each country depicted in (S1 Table). The mean age and standard deviation of these women were 35.8 ±7.8 respectively. Regarding education, has no formal education 91,068 (38.7%), only primary education 98,071 (41.7%), women achieving secondary 36,095 (15.4%) and higher education 9,852 (4.19%). Occupational status shows that 156,101 women (66.4%) are employed, while 78,984 (33.6%) are not. Partner education levels reveal that 71,839 (36.7%) have no education and 77,997 (39.8%) have only primary education. Media exposure is nearly evenly split, with 118,933 women (50.6%) having no exposure and 116,153 (49.4%) having exposure. Antenatal care (ANC) visits indicate that 30,942 women (94.77%) attended ANC, while only 1,708 (5.23%) did not. Wealth distribution shows that 52,599 women (22.4%) belong to the poorest category, while 39,479 (16.8%) are in the richest category. Community-level maternal literacy is low for 186,618 women (79.4%), and community wealth is predominantly low income (160,658; 68%). In terms of residency, 72,230 women (30.7%) live in urban areas, while 162,855 (69.3%) reside in rural areas. Country-wise, Tanzania has the highest representation at 80,373 (34.2%), followed by Kenya (70,854; 30.1%) and Burkina Faso (48,193; 20.5%) (Table 1).

### Numbers of pregnancy loss among ever-pregnant women

The total number of pregnancies lost among ever-pregnant women in SSA is 44,877. The maximum number of pregnancy loss per woman is 9 among ever-pregnant women in SSA. The median number of pregnancy losses among ever-pregnant women is 2.67±0.63 with a 95% confidence interval of (2.64, 2.69) with a median number of abortions 7.9 (7.6, 8.1), miscarriage 6.3 (6.1, 6.4), and stillbirth 2.3 (2.2, 2.5) (Table 2).

### Random effects (measures of variation)

The number of pregnancy loss among women who have been ever pregnant varies significantly across each cluster. Intra-cluster correction coefficient indicated that 11.3% of the variation in the number of pregnancy losses among women who have been ever pregnant was attributed to community-level factors. Proportional change of variability in the final model showed that 42.8% of the variation in the number of pregnancy losses among women who have been ever pregnant in Sub-Saharan African countries was explained across communities or countries ((Table 3).

Key: AIC: Akakian information Criteria, BIC: Bayesian information Criteria, ICC: Intra-class correlation, PCV: Proportional Change of Variability. The first model was the model-0= empty model or null model was conducted without an independent variable (univariate analysis) and the result showed intra-class correlation (ICC) = 11.3%, the second model was model 1= analyzing only individual-level variable, the 3rd model was model-2 (analyzing only community-level variable), the last model was model 3= analyzing both individual and community-level variable.

### Predictors of the number of pregnancy loss among ever-pregnant women in Sub-Saharan Africa using multilevel mixed effect negative binomial regression

Different count models were fitted for model selection and the multilevel mixed effect negative binomial regression was selected over the standard Poisson, negative binomial, and zero-inflated regression models using ICC value. The final multilevel mixed effect negative binomial regression model was selected based on low AIC value (AIC=27,457). In this study, maternal age, maternal education, partner education, number of under-five children, and country of residence were statistically significant at p-value <0.05.

The number of pregnancy loss increased by 6.7% as the one-year age of the mother was increased [AIRR =1.067, 95%CI (1.06, 1.07)]. The incidence risk ratio of pregnancy loss is high among ever-pregnant women who have primary education [AIRR= 1.10, 95% CI (1.01, 1.22)] as compared to women who have no education. The incidence risk ratio of pregnancy loss is high among ever-pregnant women with partners who have higher education [AIRR=1.18, 95% CI (1.04, 1.39)] as compared with women with partners who have no education. The incidence risk ratio of pregnancy loss is high among women from households with more household members [AIRR= 1.02, 95% CI (1.01, 1.03)]. The incidence risk ratio of pregnancy loss is less likely among ever-pregnant women who have a higher number of under-five children [AIRR =0.95, 95% CI (0.91, 0.99)]. The incidence risk ratio of pregnancy loss is more likely among ever pregnant women who have been in Cote'divore [AIRR =1.76, 95% CI (1.6, 2)] than ever pregnant women who have been in Burkina Faso (Table 4).

## Discussion

According to the WHO, maternal health problems related to pregnancy are a public health issue globally. Moreover, the issue is more pronounced in underdeveloped regions

**Table 1.  Socio-demographic characteristics ever-pregnant women in Sub-Saharan Africa using the latest DHS from 2015-2023 (weighted N = 235,086).**

| Characteristics | Category | Frequency | Percentage |
|---|---|---|---|
| Maternal educational status | No education | 91068 | 38.7 |
|  | Primary | 98071 | 41.7 |
|  | Secondary | 36095 | 15.4 |
|  | Higher | 9852 | 4.19 |
| Maternal occupation | No | 78,984 | 33.6 |
|  | Yes | 156,101 | 66.4 |
| Paternal education | No education | 71,839 | 36.7 |
|  | Primary education | 77,997 | 39.8 |
|  | Secondary education | 33,538 | 17.1 |
|  | Higher education | 12,526 | 6.4 |
| Media exposure | No | 118,933 | 50.6 |
|  | Yes | 116,153 | 49.4 |
| ANC visit | No | 1,708 | 5.23 |
|  | Yes | 30,942 | 94.77 |
| Wealth index | Poorest | 52,599 | 22.4 |
|  | Poorer | 49,975 | 21.3 |
|  | Middle | 48,267 | 20.5 |
|  | Richer | 44,765 | 19.0 |
|  | Richest | 39,479 | 16.8 |
| Community level maternal literacy | Low | 186,618 | 79.4 |
|  | High | 48,467 | 20.6 |
| Community level wealth index | Low income | 160,658 | 68 |
|  | High income | 74,427 | 32 |
| Community level maternal media exposure | Low | 170,747 | 72.6 |
|  | Higher | 64,339 | 27.4 |
| community-level maternal ANC utilization | Low | 142,971 | 60.9 |
|  | High | 91,953 | 39.1 |
| Residence | Urban | 72,230 | 30.7 |
|  | Rural | 162,855 | 69.3 |
| Country | Burkina Faso | 48,193 | 20.5 |
|  | Cote'divore | 35,665 | 15.2 |
|  | Kenya | 70,854 | 30.1 |
|  | Tanzania | 80,373 | 34.2 |

Key, ANC= antenatal care

**Table 2.  Number of pregnancy loss among ever-pregnant women in SSA from 2015-2023.**

| Pregnancy losses | Number of pregnancy loss |
|---|---|
| Median number of pregnancy loss | 2.67(2.64, 2.69) |
| Minimum value | 0 |
| Maximum value | 9 |
| Median number of abortion | 7.9(7.6-8.1) |
| Median number of miscarriage | 6.3(6.1-6.4) |
| Median number of stillbirths | 2.3(2.2-2.5) |

**Table 3. Model building, model selection, and cluster variation.**

| Criteria | Model 0 | Model 1 | Model 2 | Model 3 |
|---|---|---|---|---|
| Variance of each model | 0.42 | 0.24 | 0.40 | 0.21 |
| ICC | 11.3% | 6.8% | 10.8% | 6.77% |
| PCV | Reference | 42.4% | 4.7% | 42.8% |
| AIC | 296,074 | 27,620 | 29,4528 | 27,457 |
| BIC | 296,105 | 27,786 | 294,642 | 270,689 |

like SSA. Therefore, the objective of the study was to estimate the number of pregnancy losses and identify predictors among ever-pregnant women in SSA using recent rounds of DHS data. The study found that the pregnancy loss among ever-pregnant women is approximately 3 in SSA. This study finding is higher than the median number of pregnancy losses reported in the United States [9], Manitoba [28], and the United Kingdom [29]. This highlights a significant public health issue regarding pregnancy loss in SSA compared to more developed regions. The disparity in the rates of pregnancy loss may be due to limited access to healthcare, socioeconomic disparities, and cultural practices that may affect maternal health, lack of access to skilled healthcare providers, and insufficient maternal health education in SSA. This finding highlights the need for better maternal healthcare in SSA.

The study identified predictors that are associated with pregnancy loss, including maternal age, maternal education, partner education, household size, number of under-five children, and country. With a one-year increase in the age of the mother, the number of pregnancy losses increases by 6.7%. The study conducted in Ghana [19] supports this study's findings. The risk of pregnancy loss increases as the age of the mother increases. This may be because as a mother's age increases they may have chronic health problems intern that can also increase the risk of pregnancy loss.

Women with primary education have a higher incidence risk of pregnancy loss compared to women with no education. This finding was contradicted by the previous study findings [19,30,31]. Women with no education may have less access to healthcare (like induced abortion), which could lead to a lower incidence risk of pregnancy loss and women. However, this finding is not supported by other study findings [17,18]. The logical reason may be due to educated mothers may have delayed childbearing and pregnancy due to waste of their time in education and learning which in turn increases chromosomal abnormality in fetes which increases pregnancy loss.

Women with partners in higher education have a higher risk of pregnancy loss. However, this finding is not supported by other study findings [17,18]. Higher educational status is linked to pregnancy delays in both parents and late pregnancy intern increases chromosomal abnormality in fetes which increases pregnancy loss.

Women with a higher number of household members have a higher incidence risk of pregnancy loss. These findings may reflect the challenges faced by women in larger households in terms of resource allocation, maternal healthcare utilization, and the ability to provide adequate care for existing pregnancies, which can impact pregnancy loss [32].

Women with a higher number of under-five children are less likely to experience pregnancy loss. The higher number of under-five children may reflect better obstetric history, timely pregnancy initiation, and access to healthcare and thus prevent chromosomal problems that lead to pregnancy loss [20,21]. The other possible explanation could

**Table 4. Multilevel mixed effect negative binomial regression analysis for predictors of the number of pregnancy loss among ever-pregnant women in SSA using the latest DHS data from 2015-2023 (weighted N=235,086).**

| Characters | Model 0 | Model 1 | Model 2 | Model 3 |
|---|---|---|---|---|
| | ICC=11.3% | AIRR (95%CI) | AIRR (95%CI) | AIRR (95% CI) |
| Maternal age | | 1.06(1.05,1.07) | | 1.067(1.0,1.07)* |
| Maternal education | | | | |
| No education | 1 | | | |
| Primary | | 1.0(0.9,1.1) | | 1.10(1.01,1.22)* |
| Secondary | | 0.9(0.8,1.0) | | 1.06(0.94,1.19) |
| Higher | | 0.8(0.7,0.9) | | 0.94(0.78,1.14) |
| Working status | | | | |
| No | 1 | | | |
| Yes | | 1.04(0.97,1.12) | | 1.05(0.98,1.12) |
| Partners' Education | | | | |
| No education | 1 | | | |
| Primary | | 0.98(0.89,1.07) | | 1.08(0.98,1.19) |
| Secondary | | 1.03(0.92,1.16) | | 1.07(0.96,1.21) |
| Higher education | | 1.17(1.00,1.38) | | 1.18(1.01,1.39)* |
| Media exposure | | | | |
| No | | 1 | | |
| Yes | | 0.11(0.03,1.21) | | 0.93(0.92,1.08) |
| Number of ANC visit | | 1.09(1.07,1.11) | | 1.09(0.97,1.11) |
| Number of household members | | 1.01(1.00,1.03) | | 1.02(1.01,1.03)* |
| Number of under-five children | | 0.95(0.91,0.99) | | 0.95(0.91,0.99)* |
| Age of household head | | 1.00(0.99,1.00) | | 1.00(0.99,1.00) |
| Household wealth index | | | | |
| Poorest | | 1 | | |
| Poorer | | 0.98(0.88,1.09) | | 0.98(0.88,1.09) |
| Middle | | 0.94(0.84,1.05) | | 0.96(0.85,1.07) |
| Richer | | 1.04(0.92,1.17) | | 1.09(0.96,1.24) |
| Richest | | 1.04(0.91,1.18) | | 1.12(0.96,1.31) |
| Place of residence | | | | |
| Urban | | 1 | | |
| Rural | | | 0.77(0.75,0.80) | 0.99(0.90,1.10) |
| Community-level maternal literacy | | | | |
| Low | | 1 | | |
| High | | | 0.87(0.79,0.96) | 0.92(0.79,1.07) |
| Community-level wealth index | | | | |
| A low proportion of rich | | 1 | | |
| A high proportion of rich | | | 0.89(0.82,0.97) | 0.99(0.89,1.11) |
| Community-level maternal media exposure | | | | |
| A low proportion of media exposure | | 1 | | |
| The high proportion of media exposure | | | 1.02(0.92,1.12) | 0.98(0.86,1.12) |
| Community-level maternal ANC | | | | |
| A low proportion of ANC utilization | | 1 | | |
| The high proportion of ANC utilization | | | 0.98(0.90,1.06) | 0.96(0.87,1.07) |
| Country | | | | |
| Burkina Faso | | 1 | | |

*(Continued)*

**Table 4.** (Continued)

| Characters | Model 0 | Model 1 | Model 2 | Model 3 |
|---|---|---|---|---|
| | ICC=11.3% | AIRR (95%CI) | AIRR (95%CI) | AIRR (95% CI) |
| Cote'divore | | | 1.49(1.44,1.54) | 1.79(1.60,2.00)* |
| Kenya | | | 0.99(0.95,1.03) | 1.05(0.91,1.21) |
| Tanzania | | | 1.04(1.01,1.07) | 0.94(0.84,1.06) |

Key: 1= Reference category, * =significant at p-value <0.05 ANC= Antenatal care, # = number, AIRR= Adjusted incidence rate ratio.

Note: The first model was the model-0= empty model or null model was conducted without an independent variable (univariate analysis) and the result showed intra-class correlation (ICC) = 61.5%, the second model was model 1= analyzing only individual-level variable, the 3rd model was model-2 (analyzing only community-level variable), the last model was model 3= analyzing both individual and community-level variable.

women who had repeated pregnancies with normal births may have gained experience and knowledge about pregnancy care and healthy behaviors, leading to better management of their pregnancies and a lower risk of pregnancy loss. Women who ever pregnant in Cote'divore have a higher incidence risk of pregnancy loss. This evidence is supported study conducted in Cote'divore [33]. This research findings indicate that women in Côte d'Ivoire experience a higher incidence of pregnancy loss, it underscores the need for attention to this issue within the context of maternal health which highlights the need for country-specific strategies.

## Strength and limitation

The strength of this study is the use of nationally representative data, which allows it to be generalizable, and the use of the count model, which allows it to estimate the number of pregnancy losses and its predictors among women who have ever been pregnant. The DHS surveys rely on self-reported information, which can be subject to recall bias or social desirability bias, and since this study included only four countries that have data related to pregnancy loss (abortion, miscarriage, and stillbirth) may not be representative for SSA countries were may be the limitation of this study.

## Conclusion

The findings indicate that, on average, there are three pregnancy losses among ever-pregnant women in Sub-Saharan Africa. Notably, a one-year increase in maternal age and higher education levels for both mothers and their partners are linked to an increased risk of pregnancy loss. In contrast, mothers with multiple children generally experience lower rates of loss. Therefore, policy interventions should address the heightened risk of pregnancy loss linked to advanced maternal age and higher education levels for both mothers and their partners. This can be achieved by supporting programs that educate prospective parents about the effects of maternal age on pregnancy outcomes. Furthermore, promoting flexible educational pathways and providing career support can encourage healthier timing for pregnancies. Additionally, initiatives that support families and promote larger family sizes may help reduce pregnancy loss rates in Sub-Saharan Africa.

## Supporting information

**S1 Table. Socio-demographic characteristics among ever-pregnant women in Sub-Saharan Africa using the latest DHS 2015-2023 (weighted frequency).**
(DOCX)

## Acknowledgments

The authors are sincerely grateful to the DHS program for providing us to use the DHS dataset through their archives https://dhsprogram.com/

## Author contributions

**Conceptualization:** Abel Endawkie.

**Data curation:** Abel Endawkie.

**Formal analysis:** Abel Endawkie.

**Investigation:** Abel Endawkie.

**Methodology:** Abel Endawkie, Yawkal Tsega.

**Software:** Abel Endawkie.

**Supervision:** Abel Endawkie.

**Validation:** Abel Endawkie, Yawkal Tsega.

**Visualization:** Abel Endawkie.

**Writing – original draft:** Abel Endawkie, Yawkal Tsega.

**Writing – review & editing:** Abel Endawkie, Yawkal Tsega.

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
