## [Decision Letter · Decision Letter 0]

7 Nov 2024

PGPH-D-24-01592

Pregnancy Loss and Its Predictors among Women Who Ever Pregnant in Four Sub-Saharan African Countries: Multilevel Mixed Effect Negative Binomial Regression

Dear Dr. Endawkie,

Thank you for submitting your manuscript to PLOS Global Public Health. After careful consideration, we feel that it has merit but does not fully meet PLOS Global Public Health’s publication criteria as it currently stands. Therefore, we invite you to submit a revised version of the manuscript that addresses the points raised during the review process.

Please note that we have only been able to secure a single reviewer to assess your manuscript. We are issuing a decision on your manuscript at this point to prevent further delays in the evaluation of your manuscript. Please be aware that the editor who handles your revised manuscript might find it necessary to invite additional reviewers to assess this work once the revised manuscript is submitted. However, we will aim to proceed on the basis of this single review if possible. 

The reviewer has raised a number of major concerns, please could you carefully address all comments raised? 

We look forward to receiving your revised manuscript.

Kind regards,

Johanna Pruller, Ph.D.

PLOS Staff Editor

Journal Requirements:

1. Thank you for uploading your study's underlying data set. Unfortunately, the repository you have noted in your Data Availability statement does not qualify as an acceptable data repository according to PLOS's standards.

2. Please provide an Author Summary. This should appear in your manuscript between the Abstract (if applicable) and the Introduction, and should be 150–200 words long. The aim should be to make your findings accessible to a wide audience that includes both scientists and non-scientists. Sample summaries can be found on our website under Submission Guidelines: 

https://journals.plos.org/globalpublichealth/s/submission-guidelines#loc-parts-of-a-submission

Additional Editor Comments (if provided):

Reviewers' comments:

Reviewer's Responses to Questions

**Comments to the Author**

1. Does this manuscript meet PLOS Global Public Health’s publication criteria ? Is the manuscript technically sound, and do the data support the conclusions? The manuscript must describe methodologically and ethically rigorous research with conclusions that are appropriately drawn based on the data presented.

Reviewer #1: Partly

2. Has the statistical analysis been performed appropriately and rigorously?

Reviewer #1: Yes

3. Have the authors made all data underlying the findings in their manuscript fully available (please refer to the Data Availability Statement at the start of the manuscript PDF file)?

Reviewer #1: Yes

4. Is the manuscript presented in an intelligible fashion and written in standard English?

Reviewer #1: Yes

5. Review Comments to the Author

Reviewer #1: Dear Authors,

This is a good work and a lot of effort was put into this document to ensure that it meets PloS Global Public Health. That said, i have highlighted some comments in the attached document that will help in my view improve the outlook of the manuscript.

6. PLOS authors have the option to publish the peer review history of their article (what does this mean? ). If published, this will include your full peer review and any attached files.

**Do you want your identity to be public for this peer review?** For information about this choice, including consent withdrawal, please see our Privacy Policy .

Reviewer #1: **Yes: ** Bwalya Bupe Bwalya

---

## [Decision Letter · Decision Letter 1]

10 Dec 2024

PGPH-D-24-01592R1

Number of Pregnancy Loss and Its Predictors among Ever Pregnant Women in Sub-Saharan Africa: Multilevel Mixed Effect Negative Binomial Regression

Dear Dr. Endawkie,

Thank you for submitting your manuscript to PLOS Global Public Health. After careful consideration, we feel that it has merit but does not fully meet PLOS Global Public Health’s publication criteria as it currently stands. Therefore, we invite you to submit a revised version of the manuscript that addresses the points raised during the review process.

We look forward to receiving your revised manuscript.

Kind regards,

Mohammad Shahidul Islam, PhD

Academic Editor

Journal Requirements:

Additional Editor Comments (if provided):

I have read the manuscript by Endawkie et al. titled “Number of Pregnancy Loss and Its Predictors among Ever Pregnant Women in Sub-Saharan Africa: Multilevel Mixed-Effect Negative Binomial Regression” with great interest. I agree with the authors that understanding the underlying causes of pregnancy loss is crucial for reducing uneventful pregnancies, which often bring significant social stigma to families and increase their vulnerability. Sub-Saharan Africa, in particular, lags behind other regions in addressing this issue, underscoring the importance of this study.

I have also reviewed how the authors addressed the flagged areas raised by previous reviewers. While the current version of the manuscript resolves most of these concerns, there are still some areas requiring improvement before it is ready for publication in PLOS Global Public Health Journal. Below are my specific comments and suggestions for revisions:

Abstract

• Line 20: Please specify whether the value 0.267 represents the mean pregnancy loss per woman or the proportion of pregnancy loss.

• Line 23: Clarify what is meant by "a higher number of under-five children." Does this refer to a higher count of children under five per household or something else?

• Lines 26–32: The conclusion is somewhat repetitive of the results section. Consider providing a concise take-home message with actionable steps or policy recommendations to reduce pregnancy loss risk in sub-Saharan Africa (SSA).

Introduction

• Line 68: The phrase "information loss" is ambiguous. Does this refer to a lack of data on pregnancy loss? Please clarify.

Methods

• Line 79: The sentence implies that the authors conducted the survey themselves. Please revise so readers understand that DHS data was used.

• Line 92: The term "viable birth" may need clarification. Consider replacing it with pregnancy outcome for greater accuracy.

Results

• Line 268: The authors express the value in percentages here, while using proportion in other sections. Please consider using a consistent format throughout for better readability.

• Table 4: Including a p-value for the adjusted incidence rate ratio (AIRR) would facilitate easier interpretation.

• A supplementary table showing the population characteristics by country would enhance the manuscript, as country-specific variations in population characteristics could influence the overall outcomes.

Discussion

• Line 293: The format of presenting values (e.g., "3 in 10 pregnancies") is inconsistent with other sections where mean numbers are used. Please adopt a single format (either proportion or percentage) throughout the manuscript.

• Line 304: The statement about mean pregnancy loss in Burkina Faso being relatively higher is unclear. Specify what it is being compared to and include appropriate citations.

• Line 315: The claim that primary education is linked with increased pregnancy loss is not supported by the data presented in Table 4. Please provide additional evidence or clarify this point.

---

## [Editor Report · Decision Letter 2]

6 Jan 2025

PGPH-D-24-01592R2

Number of Pregnancy Loss and Its Predictors among Ever Pregnant Women in Sub-Saharan Africa: Multilevel Mixed Effect Negative Binomial Regression

Dear Dr. Endawkie,

Thank you for submitting your manuscript to PLOS Global Public Health. After careful consideration, we feel that it has merit but does not fully meet PLOS Global Public Health’s publication criteria as it currently stands. Therefore, we invite you to submit a revised version of the manuscript that addresses the points raised during the review process.

We look forward to receiving your revised manuscript.

Kind regards,

Mohammad Shahidul Islam, PhD

Academic Editor

Journal Requirements:

Additional Editor Comments:

Thank you for sharing the revised version of the manuscript titled “Number of Pregnancy Loss and Its Predictors among Ever Pregnant Women in Sub-Saharan Africa: Multilevel Mixed-Effect Negative Binomial Regression.” The revisions have enhanced the clarity regarding the study’s methodology and findings. However, I still have a few concerns:

• Line 25: The value 0.267 does not fully align with the explanation provided in the feedback. In the results section, lines 240-242, the authors reported “average number of pregnancy losses among ever-pregnant women was 2.67±0.63 per 10 pregnancies.” Hence, the value 0.267 reflects a proportion rather than the mean number of pregnancy losses per woman. Please clarify.

• I appreciate the effort to clarify the term "information loss" in Line 76, but the statement remains unclear. I understand the authors' point that treating women with multiple pregnancy losses the same as those with a single loss may obscure important distinctions. I suggest simplifying this phrasing to enhance readability.

• The title "Mean numbers of pregnancy loss among ever-pregnant women" (line 239) seems inappropriate as the data presented in the manuscript is expressed as a proportion. Additionally, interpreting findings based on the average number of pregnancy losses per woman does not make sense, as it is expected that older women will have a higher mean number of pregnancy losses than younger women, since younger women have experienced fewer pregnancies and therefore face less risk of pregnancy loss.

• The manuscript suggests primary education is linked to increased pregnancy loss, while higher education is not. This contradicts existing literature indicating that maternal education generally improves pregnancy outcomes. The conclusion seems to rely on a 95% CI (1.01, 1.22) that narrowly excludes the null. Additionally, data from Table 1 and the univariate analysis in Table 4 do not strongly support this claim. Given the policy implications, I recommend providing stronger evidence or further clarification.

• As mentioned earlier, throughout the manuscript, the data are presented in an inconsistent format. For example, in lines 228-230, it is mentioned that the mean number of pregnancy losses was 0.28 per woman who was not educated, while in lines 240-242, it states that an average of 2.67±0.63 pregnancies were lost per 10 pregnancies. To improve readability and avoid confusion, I recommend ensuring consistent data presentation throughout the manuscript.

---

## [Editor Report · Decision Letter 3]

14 Jan 2025

PGPH-D-24-01592R3

Pregnancy Loss and Its Predictors among Ever Pregnant Women in Sub-Saharan Africa: Multilevel Mixed Effect Negative Binomial Regression

Dear Dr. Endawkie,

Thank you for submitting your manuscript to PLOS Global Public Health. After careful consideration, we feel that it has merit but does not fully meet PLOS Global Public Health’s publication criteria as it currently stands. Therefore, we invite you to submit a revised version of the manuscript that addresses the points raised during the review process.

We look forward to receiving your revised manuscript.

Kind regards,

Mohammad Shahidul Islam, PhD

Academic Editor

Journal Requirements:

Additional Editor Comments (if provided):

I thank the authors for addressing the comments. The revised manuscript addresses most of the flagged areas. However, I have noticed some inconsistencies between the text and Table 2:

1. In line 245, the median number of abortions is reported as 7.9 (7.6, 8.1), but Table 2 presents the mean. Similar inconsistencies are observed for miscarriage and stillbirth.

2. In lines 30 and 353, it is stated that there are three pregnancy losses for every ten pregnancies among ever-pregnant women in Sub-Saharan Africa. However, Table 2 indicates a median of 2.67 pregnancy losses per woman. I could not find any data in the results section supporting the claim of 3 out of 10 pregnancies being lost.

o I suggest revising this statement to indicate "three pregnancy losses per woman," or adding a row in Table 2 showing the percentage of pregnancies lost to support the claims in lines 30 and 353.

3. Please include the total number of pregnancies analyzed and the total number of pregnancy losses in Table 2 or in text for clarity.

4. The column for percentages in Table 2 appears unnecessary and could be removed.

---

## [Editor Report · Decision Letter 4]

16 Jan 2025

PGPH-D-24-01592R4

Pregnancy Loss and Its Predictors among Ever Pregnant Women in Sub-Saharan Africa: Multilevel Mixed Effect Negative Binomial Regression

Dear Dr. Endawkie, Thank you for submitting your revised manuscript to PLOS Global Public Health. I have one final comments before approving this manuscript. Since the total number of pregnancy losses is not reported in Table 2 or elsewhere, it is difficult to understand how the median number of pregnancy losses is less than the median number of abortions or miscarriages, as the latter should be subsets of pregnancy loss.

Kind regards,

Mohammad Shahidul Islam, PhD

Academic Editor

---

## [Editor Report · Decision Letter 5]

3 Feb 2025

Pregnancy Loss and Its Predictors among Ever Pregnant Women in Sub-Saharan Africa: Multilevel Mixed Effect Negative Binomial Regression

PGPH-D-24-01592R5

Dear Dr. Endawkie,

We are pleased to inform you that your manuscript 'Pregnancy Loss and Its Predictors among Ever Pregnant Women in Sub-Saharan Africa: Multilevel Mixed Effect Negative Binomial Regression' has been provisionally accepted for publication in PLOS Global Public Health.

Best regards,

Mohammad Shahidul Islam, PhD

Academic Editor
